# Kinase Inhibitor Screening Displayed *ALK* as a Possible Therapeutic Biomarker for Gastric Cancer

**DOI:** 10.3390/pharmaceutics14091841

**Published:** 2022-09-01

**Authors:** Felipe Pantoja Mesquita, Pedro Filho Noronha Souza, Emerson Lucena da Silva, Luina Benevides Lima, Lais Lacerda Brasil de Oliveira, Caroline Aquino Moreira-Nunes, William J. Zuercher, Rommel Mario Rodríguez Burbano, Maria Elisabete Amaral de Moraes, Raquel Carvalho Montenegro

**Affiliations:** 1Department of Medicine, Pharmacogenetics Laboratory, Drug Research and Development Center (NPDM), Federal University of Ceará, Fortaleza 60430-275, CE, Brazil; 2Department of Biochemistry and Molecular Biology, Federal University of Ceará, Fortaleza 60430-275, CE, Brazil; 3Department of Biological Sciences, Oncology Research Center, Federal University of Pará, Belém 66073-005, PA, Brazil; 4Division of Chemical Biology and Medicinal Chemistry, University of North Carolina at Chapel Hill, Chapel Hill, NC 27599, USA; 5Laboratory of Human Cytogenetics, Institute of Biological Sciences, Federal University of Pará, Belém 66073-005, PA, Brazil; 6Molecular Biology Laboratory, Ophir Loyola Hospital, Belém 66073-005, PA, Brazil

**Keywords:** gastric cancer, kinase inhibitor, *ALK*, targeted therapy

## Abstract

Despite advances in cancer chemotherapy, gastric cancer (GC) continues to have high recurrence rates and poor prognosis with limited treatment options. Understanding the etiology of GC and developing more effective, less harmful therapeutic approaches are vital and urgent. Therefore, this work describes a novel kinase target in malignant gastric cells as a potential therapeutic strategy. Our results demonstrate that among 147 kinase inhibitors (KI), only three molecules were significantly cytotoxic for the AGP-01 cell line. Hence, these three molecules were further characterized in their cellular mode of action. There was significant cell cycle impairment due to the expression modulation of genes such as *TP53*, *CDKN1A*, *CDC25A*, *MYC*, and *CDK2* with subsequent induction of apoptosis. In fact, the Gene Ontology analysis revealed a significant enrichment of pathways related to cell cycle regulation (GO:1902749 and GO:1903047). Moreover, the three selected KIs significantly reduced cell migration and Vimentin mRNA expression after treatment. Surprisingly, the three KIs share the same target, *ALK* and *INSR*, but only the *ALK* gene was found to have a high expression level in the gastric cancer cell line. Additionally, lower survival rates were observed for patients with high *ALK* expression in TCGA-STAD analysis. In summary, we hypothesize that *ALK* gene overexpression can be a promising biomarker for prognosis and therapeutic management of gastric adenocarcinoma.

## 1. Introduction

Gastric cancer (GC) is the second major cause of cancer-related deaths worldwide and the third leading cause in industrialized nations [1,2]. Despite improvements in the management of GC patients with distant metastases, significant recurrence rates and a poor prognosis persist, with limited treatment choices and a median survival around one year [3,4]. Another problem is that GC is a very complex illness with a complicated etiology involving complex host genetic and environmental variables [5,6].

Until now, only a few targeted therapeutic agents, such as trastuzumab and ramucirumab, have been approved by the US Food and Drug Administration for gastric cancer patients identified with the respective genetic defects [7]. However, only 7–34% of GC patients have the altered molecular target for these agents and therefore most GC patients must continue to depend on cytotoxic chemotherapy and/or surgical resection as the current standard of care [8]. As a result, a greater number of patients evolve with poor prognostic and low survival rate because of therapeutic failure [9,10]. Therefore, understanding the etiology of GC and the development of more effective, less hazardous therapy techniques are critical and urgent.

In this sense, protein kinases are central actors in most of signal transduction pathways. Because their function is frequently disrupted in signal transduction networks leading to illnesses like cancer and inflammation, several of the 518 protein kinases encoded by the human genome have emerged as potential therapeutic targets [11]. Nowadays, there are around 250 kinase inhibitors (KIs) in clinical studies, with 37 of them being authorized for human usage [12]. Thus, it is critical to search new compounds’ targets in depth in order to comprehend their molecular and cellular modes of action (MoA).

Thus, several kinase activity and binding screens have been published in the past few years. Among them, Elkins and co-workers reported a comprehensive characterization of 367 tool compounds screened against 224 recombinant kinases [13]. Despite the considerable value of these studies, the cellular or clinical evaluation of KIs has not yet been systematically analyzed. Therefore, the published kinase inhibitors library (PKIS) can be a useful tool for target screening in different cancer models. Taking all together, the present study aimed to evaluate possible novel kinase targets in malignant gastric cells in order to identify a new therapeutic strategy based on kinase inhibition using PKIs library.

## 2. Materials and Method

### 2.1. Cell Culture

Leal and co-workers (2009) developed a cell line from a patient with metastasis of gastric adenocarcinoma (AGP-01) as an alternative model for the anticancer drug screening model. A normal gastric mucosa cell line MNP-01 was used as normal cellular control [14,15]. Cells were cultured in sterilized flasks containing filtered Dulbecco’s modified Eagle’s medium (DMEM; Gibco^®^, New York, NY, USA) supplemented with 10% fetal bovine serum (Gibco^®^), 100 U/mL penicillin, and 100 µg/mL streptomycin. The culture flasks were maintained in a 5% CO_2_ air-humidified atmosphere at 37 °C.

### 2.2. Alamar Blue Method

The growth inhibitory activity of 147 kinase inhibitors (KI) against the AGP-01 cell line was first evaluated at 10 µM concentration (72 h of treatment) using the Alamar Blue method. The growth inhibition (%) was calculated as comparing both KIs based on the cellular viability of treated and KI-untreated AGP-01 cells. Then, relevant KIs that obtained >80% of growth inhibition were selected for IC_50_ determination by a concentration-response curve. KI screening was carried out on 384-well plate (Corning^®^, Glendale, AZ, USA). Firstly, 300 cells per well were seeded and incubated overnight at 37 °C and 5% CO_2_. Then, KIs were diluted with DMEM medium and subsequently added in the plate to achieve final desired concentration (10 µM). After 72 h of incubation, Alamar Blue reagent was added to measure cell viability. Fluorescence intensity (560/590 nm) was measured using Beckman Coulter Microplate Reader DTX 880. Cells treated with 0.1% dimethyl sulfoxide (DMSO) were used as negative control and doxorubicin (Doxo, Milan, Italy) was used as positive control.

### 2.3. Cell Cycle Progression

AGP-01 cells were seeded at 2 × 10^3^ cells per well in a 96-well plate and treated with a single concentration (1 µM) of each KI selected in the screening. After 72 h of exposure, cells were harvested, washed with PBS 1X, and fixed with ice-cold 80% ethanol for 1 h. Then, cells were incubated with RNAse A (200 µg/mL) and propidium iodide (50 µg/mL) for 30 min. DNA content was analyzed using flow cytometry (BD FACSVerse^TM^, BD Biosciences, East Rutherfor, NJ, USA). In total, 10,000 events were acquired, and data were analyzed using FlowJo^®^ software (Version 10.5.3, FlowJo LLC, Ashland, Wilmington, DE, USA).

### 2.4. Caspase 3/7 Activity

The caspase-3 and caspase-7 activities were evaluated using the CellEvent^®^ Caspase-3/7 kit according to the manufacturer’s protocol (Invitrogen^TM^, Waltham, MA, USA). AGP-01 cells were plated (3 × 10^3^ cells/well) in a 96-well plate and treated with KIs (1 µM) for 72 h. Then, cells were incubated in 5 µM of CellEvent^®^ reagent for 30 min and evaluated in Cytation^TM^ with absorption and fluorescence emission spectra of 511/533 nm. Apoptotic cells were labelled with bright fluorescent green nuclei.

### 2.5. Wound Healing Assay

The migratory potential of AGP-01 cells were evaluated by the wound healing assay as previously described by Mesquita et al. (2021) [16]. Cells were seeded (2 × 10^4^ cells/well) in a 96-well plate and scratches were made in each well with a sterile pipette tip. Then, cells were treated with KIs (1 µM) for 24 h. Pictures were taken at time 0 and 24 h from wells, and the scratch area was measured using ImageJ software (National Institutes of Health, Rockville, MD, USA).

### 2.6. Gene Expression by Quantitative Real-Time PCR

Quantitative real-time PCR (qPCR) analyses were performed with QuantStudio5 Real-Time PCR system (Applied Biosystems^®^, Waltham, MA, USA) in a 96-well PCR plate using 3 μL of cDNA obtained after reverse-transcription of total RNA (control and treated sample) using High-Capacity cDNA Reverse Transcription Kit (Applied Biosystems™), 1 μL of each primer/probe in forward and reverse, 12.5 μL of PowerUp SYBR^®^ Master Mix (Life Technologies, Carlsbad, CA, USA), and 8.5 μL of ultra-pure sterile water. The thermocycling was applied according to the manufacturer’s protocol of PowerUp SYBR^®^ Master Mix. Each assay was performed at least three times according to Minimum Information for Publication of Quantitative Real-Time PCR Experiments Guidelines [17]. The gene expression levels were based on relative analyses and calculated using the 2^−ΔΔCT^ (delta-delta threshold cycle) method [18].

### 2.7. PPI Network and GENE Ontology Enrichment

The Protein–Protein interaction (PPI) Network and Gene Ontology (GO) enrichment analyses of the differentially expressed genes were constructed using the string (https://string-db.org/ (accessed on 9 March 2022) freely available and yfile plug-in in Cytoscape software to analyze the interactions between them. False discovery rate was setup in <0.005.

### 2.8. Survival Analysis

The GEPIA database (http://gepia.cancer-pku.cn/ (accessed on 22 March 2022) was used to perform in silico correlation between gene expression and survival analyses [19]. The Cancer Genome Atlas-Stomach Adenocarcinoma (TCGA-STAD) is one of the cancer transcriptome data sets available through the GEPIA server. The database was used to compare the expression profiles of the *ALK* gene in normal and tumor tissue from gastric adenocarcinoma patients, categorizing them as high or low expression, which was then correlated with patient survival probability.

### 2.9. Statistical Analysis

All tests were performed in three independent experiments in triplicate and shown as a mean ± standard deviation (SD). The treated groups (KIs) were compared to the untreated group (DMSO) by Analysis of Variance (ANOVA) followed by Bonferroni’s posttest or by the *t*-test, considering significant differences with an interval of confidence of 95% (*p* < 0.05). GraphPad Prism 5.01 (GraphPad Software, San Diego, CA, USA) was used for data analysis and graph design.

## 3. Results

### 3.1. Screening of KIs Compounds against Metastatic Gastric Cancer Cell Line

AGP-01 cells were treated with a single concentration (10 µM) of 147 PKIS compounds to search for the most potent inhibitor based on cell growth inhibition (Figure 1). Among all, only six compounds suppressed the cell growth equally or even higher than the positive control doxorubicin (87% of growth inhibition): GW580509X, GSK1713088A, GSK1751853A, GSK2186269A, GSK1173862A, GSK2220400A (Figure 1B). Then, the non-linear regression was performed to assess the IC_50_ values of these inhibitors (Figure 1C), which confirmed that only three compounds had highly cytotoxic potential, in the micromolar range, GSK2186269A, GSK1173862A, and GSK2220400A (IC_50_ < 2.2 µM). Therefore, these three compounds were selected to proceed with further analyses.

### 3.2. Kinase Inhibitors Act as Anti-Cancer Agents by Regulating Cell Cycle and Mesenchymal Genes

The anti-proliferative effects of GSK2186269A, GSK1173862A and GSK2220400A were then evaluated by the cell cycle interaction on flow cytometry. GSK2220400A (1 µM) induced a significant G0/G1-phase arrest compared to the untreated control, while the GSK2186269A (1 µM) and GSK1173862A (1 µM) compounds triggered extensive nuclei fragmentation, considered a sub-G1 phase alteration (Figure 2A,B). Furthermore, the three compounds significantly increased caspase 3 and 7 activity, suggesting the induction of apoptosis (Figure 2C). Another cancer hallmark significantly inhibited by the selected KIs was the cell migration. All three inhibitors (1 µM) were capable of preventing the wound closure by the AGP-01 cells (Figure 2D,E).

Furthermore, after KI exposure, an alteration in gene expression involved in the cell cycle regulation, cell death, and migration was observed. In AGP-01 cells, the treatment with KIs led to the suppression of *CDK2*, *c-MYC*, *CDC25A*, and *VIM* expression while increasing *CDK1NA* expression (Figure 3A). On the other hand, GSK2220400A did not influence the expression of *CDC25A*, but it was the only drug that increased *TP53* levels. The analysis of PPI networks produced by those were involved in processes such as *cell cycle*, *mitotic processes*, *UV and radiation response*, *DNA damage process*, and *DNA metabolic process* with an FDR < 0.05 (Figure 3B,C).

### 3.3. Kinase Target Expression in the Metastatic Gastric Cancer Cells

Gene expression of AGP-01 cells and non-neoplastic gastric mucosa MNP-01 (normal cell) revealed that the *ALK* gene is overexpressed in the gastric cancer cells when compared to the normal cell line. As show in Figure 4A, among the three kinase identified as the target for the three studied compounds [13], only the *ALK* gene was overexpressed in the gastric cancer cells compared to the normal cell line. Moreover, we performed the GEPIA analysis to determine the impact of high expression on patients’ survival. Gastric adenocarcinoma patients with higher expression of *ALK* gene had a lower survival probability compared to the low expression of *ALK* group (*p* = 0.0045).

## 4. Discussion

*ALK* is considered to play a key role in the nervous system’s development and function, where it regulates the basic principles of cell proliferation, survival, and differentiation in response to extracellular stimuli [20,21]. Besides chromosome rearrangements, gene overexpression has been described as the relevant abnormal alterations in the *ALK* gene in neuroblastoma, lung, and oesophageal cancer [22,23,24]. Chen et al. (2012) also showed *ALK* overexpression in non-small cell lung cancer detected by immunohistochemical techniques [25]. Another recent evidence demonstrated that the ovarian high-grade serous carcinoma (HGSC) had significantly higher cytoplasmic *ALK* expression without chromosomal rearrangement or gene alterations compared to non-HGSC ovarian carcinomas [26]. However, *ALK* overexpression in gastric adenocarcinoma has not been well investigated. To our knowledge, Fan and co-workers (2020) described the first case of gastrointestinal stromal tumor (GIST) with *ALK* overexpression; however, GIST are a different cancer identity when compared to gastric adenocarcinoma [27].

Our study aimed to screen 147 different kinase inhibitors identified and characterized by Elkins et al. (2016) in a metastatic gastric adenocarcinoma model in order to identify a possible kinase target with therapeutic relevance. After the initial screening, only six compounds suppressed the cell growth as much as the positive control and only three compounds (GSK2186269A, GSK1173862A, and GSK2220400A) continued to be investigated regarding their mode of action and to confirm their anti-cancer properties. The results were clear in demonstrating the cell cycle blockage, apoptosis induction, and the inhibitory effect on migration as well as in gene expression modulation. In fact, based on the gene expression of *CDKN1A*, *CDK2*, *CDC25A*, *TP53*, and *MYC*, the GO biological analysis revealed that the regulation of the cell cycle (GO:1902749, GO:1903047, GO:0000082, GO:1901990) and DNA damage response (GO:0034644, GO:0006977, GO:0010332) were induced after treatment with the three kinase inhibitors, which resulted in the eventual cell death [28,29,30,31]. The *VIM* gene suppression also revealed that the kinase inhibitors probably induced mesenchymal-epithelium transition which corroborated with the inhibition on cell migration [32]. A single study by [27] showed a decrease in cell viability and cell cycle progression, as well as tumor growth in a xenograft model of gastric cancer after *ALK* inhibition [33]. Therefore, our study corroborates with the possibility of the use of *ALK* target therapy against gastric cancer.

Evaluating the three compounds with high cytotoxic potency, compounds GSK2186269A, GSK2220400A, and GSK1173862A share the same targets [13], Anaplastic Lymphoma Receptor Tyrosine Kinase (*ALK*) and Insulin Receptor (*INSR*). Additionally, the GSK1173862A molecule also interacts with Dual Specificity Tyrosine Phosphorylation Regulated Kinase 1B (*DYRK1B*). These findings suggest that the gastric cancer cells probably overexpress these genes and therefore could be a relevant therapeutic target for gastric cancer management. Moreover, we performed the gene expression analysis of these targets on gastric cancer cells (AGP-01) and, interestingly, only the *ALK* gene showed a high expression level in gastric cancer cells when compared to the non-malignant cell line (sixfold changes). Looking for another evidence that *ALK* can be a biomarker for gastric cancer, we identified a lower survival rate for patients with gastric cancer with high levels of *ALK* expression in TCGA-STAD analysis [19]. Our hypothesis is that *ALK* gene overexpression can be a therapeutic target and/or a prognosis biomarker for patients with high-grade gastric adenocarcinoma. However, further studies are needed to confirm this hypothesis.

To our knowledge, only one study evaluated the *ALK* expression and this was in four Asian gastric cancer cell lines and a single Asian cohort of gastric cancer patients. *ALK* gene amplification and protein overexpression were not seen in any of these samples [34]. It is worth it to notice that gastric carcinogenesis differs between Asian and Caucasian tumors [35,36,37,38]. A few studies have described other *ALK* molecular alterations in gastric tumors which do not include gene overexpression. For example, a study showed 2.3% positive cases of *ALK* translocation by FISH using the standard criteria of at least 15% positive cells for the break-apart signal in signet ring cell carcinoma of the gastrointestinal tract [39].

Interestingly, a novel form of *ALK* gene fusion was identified, being the first gastric adenocarcinoma case with *RAB10-ALK* fusion [40]. On the contrary, another study did not find *ALK* fusions by FISH and gene sequencing methods in gastric adenocarcinoma [41]. Therefore, it seems to have a relevant difference among the population tested or genetic alteration found within *ALK* gene in those studies. Indeed, the genetic background can affect the molecular analysis of the cell line or cohorts [42]. Small-molecule *ALK* tyrosine kinase inhibitors are very effective against a group of cancers defined by chromosomal rearrangements involving the anaplastic lymphoma kinase (*ALK*) gene. The first- and second-generation of *ALK* inhibitors was designed to act by inhibiting the *ALK* chromosomal rearrangements, and currently, treatment resistance and recurring illness have gained importance [43]. Therefore, there is still a need for the discovery of new *ALK* gene-related alterations and the development of new inhibitors. There is ample evidence that *ALK* overexpression may be a promising therapeutic biomarker for other populations in different continents, which could benefit gastric cancer patients.

## 5. Conclusions

In conclusion, in vitro screening of a kinase inhibitor library reveals three potential kinase inhibitors against gastric cancer cell model. These three molecules showed significant anticancer activity by inhibiting cell proliferation and migration, as well as provoking cell death. Interestingly, the three molecules share the same targets, *ALK* and *INSR*, in which only the *ALK* gene is overexpressed in the gastric cancer cell line. Looking for a possible impact on clinical aspects, we identified that higher *ALK* expression reduces patients’ survival rate in TCGA-STAD data. We hypothesize that *ALK* gene overexpression can be a promising biomarker for gastric adenocarcinoma. However, further studies, in particular clinical studies, need to be conducted to prove this hypothesis.

## Figures and Tables

**Figure 1 pharmaceutics-14-01841-f001:**
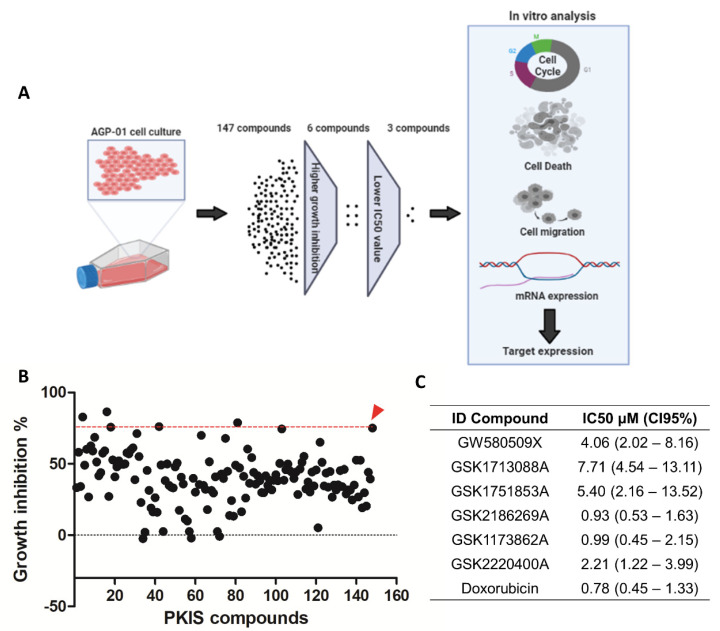
Cytotoxicity of 147 published kinase inhibitor set (PKIS) against metastatic gastric cancer cells (AGP-01). (**A**) Experimental design. After exposure to 147 published kinase inhibitors set (PKIS), only three kinase inhibitors were selected for further analyses based on their highly cytotoxic effect (lowest IC_50_ values). (**B**) Growth inhibition normalized by the untreated control (DMSO) of 147 kinase inhibitors after 72 h of treatment. The red head arrow represents the positive control doxorubicin. (**C**) IC_50_ values obtained from non-linear regression of curve concentration-response for the kinase with IC_50_ values higher than doxorubicin.

**Figure 2 pharmaceutics-14-01841-f002:**
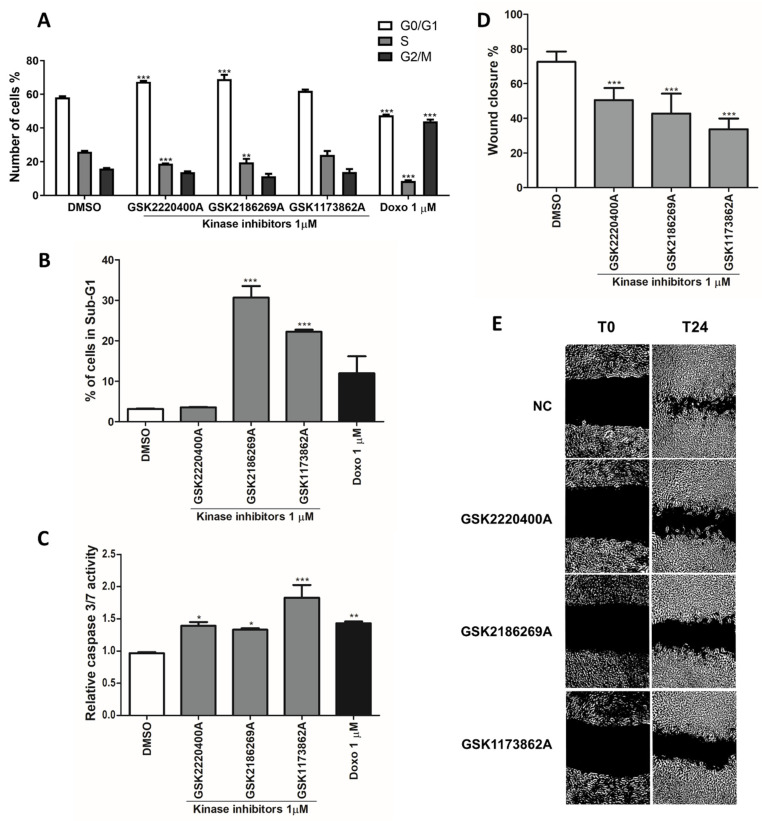
Anti-cancer effects of the three selected kinase inhibitors. (**A**) Cell cycle phase distribution of metastatic gastric cancer cells after 72 h of kinase inhibitors treatment. (**B**) Percentage of cells in sub-G1 phase after kinase inhibitors treatment. (**C**) Caspase 3 and 7 activities after 72 h of 1 µM kinase inhibitors exposure against metastatic gastric cancer cells. (**D**,**E**) Inhibitory effect on cell migration after 24 h of kinase inhibitors treatment. Mean ± standard deviation. Significant differences compared to untreated control: * *p* < 0.05, ** *p* < 0.01, *** *p* < 0.001.

**Figure 3 pharmaceutics-14-01841-f003:**
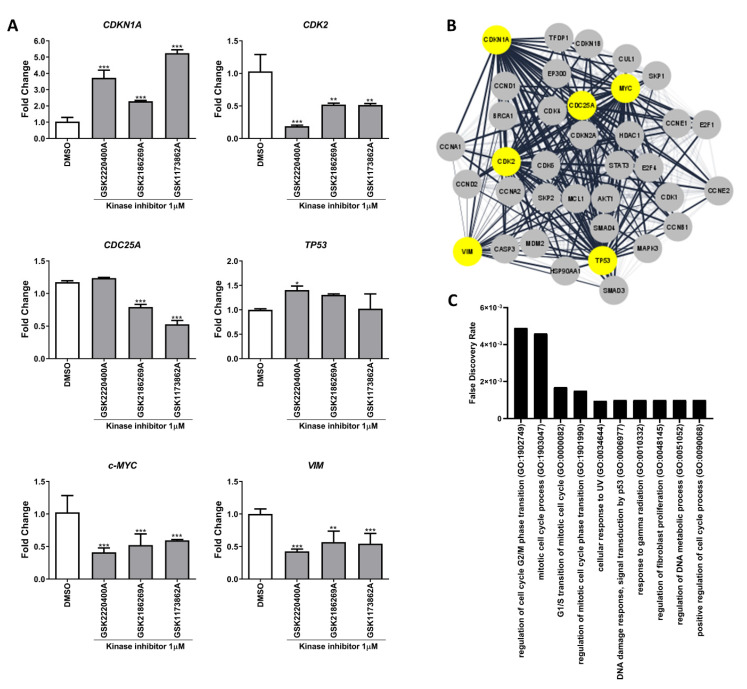
Gene expression pattern of oncogenes and tumor suppressors genes in metastatic gastric cancer cells after kinase inhibitors treatment (1 µM). (**A**) Mean ± standard deviation of mRNA expression fold change of *CDK1NA*, *CDK2*, *CDC25A*, *TP53*, *c-MYC,* and *VIM* after KI exposure. (**B**) Protein–protein interaction network analysis of the modulated genes after kinase inhibitor treatment. Analyzed genes by RT-qPCR after treatment are represented with yellow color. (**C**) False Discovery Rate (FDR) of functional analysis with GO biological processes gene set. The top 10 GO processes are sorted by the FDR. Significant differences compared to the untreated control: * *p* < 0.05, ** *p* < 0.01, *** *p* < 0.001.

**Figure 4 pharmaceutics-14-01841-f004:**
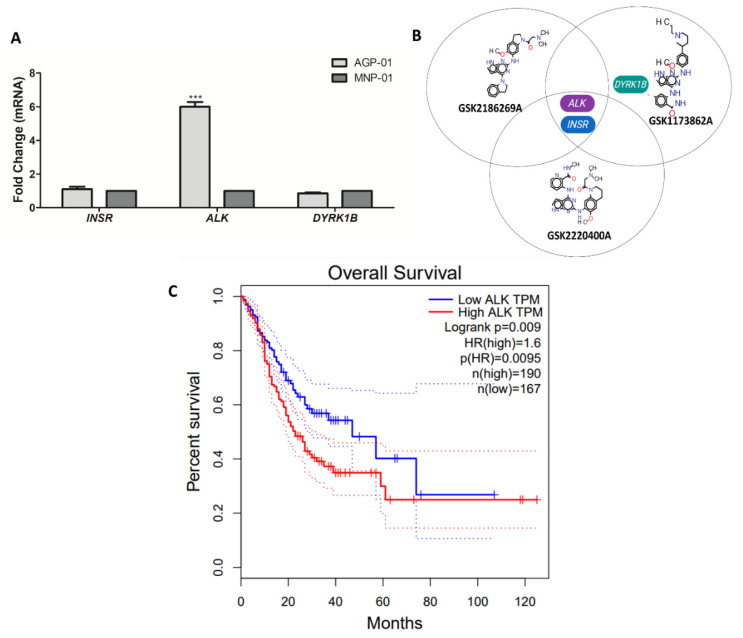
Overexpression of the *ALK* gene in the metastatic gastric cancer cells AGP-01. (**A**) Gene expression of the three-target kinase for the three selected inhibitors were assessed by RT-qPCR. The comparative expression between AGP-01 cells (metastatic gastric cancer cell) and MNP-01 cells (non-malignant gastric mucosa) are shown as mean ± standard deviation (SD). (**B**) Diagram showing the three molecules selected in this study and their targets. (**C**) GEPIA server analysis for *ALK* expression on Cancer Genome Atlas Stomach Adenocarcinoma (TCGA-STAD) comparing tumor samples and normal samples. Red and blue dotted lines represent high and low *ALK* expression, respectively. Significant differences compared to untreated control: *** *p* < 0.001.

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
