# Peer review of "Kinase Inhibitor Screening Displayed ALK as a Possible Therapeutic Biomarker for Gastric Cancer"

_pharmaceutics, 2022, doi:10.3390/pharmaceutics14091841_

Round 1
Reviewer 1 Report
The manuscript of the author describes the ALK gene overexpression can be a promising biomarker for prognosis and therapeutic management of gastric adenocarcinoma. The study is quite relevant and the references chosen are adequate. However, several issues need to be addressed:
1. In vitro screening of a kinase inhibitors library, the author reveals three potential kinase inhibitors against gastric cancer cell model, and these three molecules showed significant anticancer activity by inhibiting cell proliferation and migration, as well as provoking cell death. But there is a doubt that do these compounds actually inhibit ALK in cells, relevant evidence should be required, e,g, western blot.
2. The format of references cited by authors is inconsistent. For example, references 30, 41, 43, and so on. Although this is a small problem, it also shows the author's carelessness.
3. The clarity of all the pictures in the manuscript needs to be improved.
I would suggest to publish this paper after these questions have been addressed.
Author Response
Dear reviewer, my co-authors and I would like to thank you for the suggestions made during this high-quality review and then we present the answers to the questions.
We inform that with the reviews and suggestions, we were able to improve the idea presented by our work and we appreciate the opportunity. We hope this review has left the article suitable for publication in this high-impact journal and respect in the area.
Kind Regards.
Response to reviewer 1
The manuscript of the author describes the ALK gene overexpression can be a promising biomarker for prognosis and therapeutic management of gastric adenocarcinoma. The study is quite relevant, and the references chosen are adequate. However, several issues need to be addressed:
- In vitroscreening of a kinase inhibitors library, the author reveals three potential kinase inhibitors against gastric cancer cell model, and these three molecules showed significant anticancer activity by inhibiting cell proliferation and migration, as well as provoking cell death. But there is a doubt that do these compounds actually inhibit ALK in cells, relevant evidence should be required, e,g, western blot.
Response: Thank you for the comments. The kinase inhibitor compounds were kindly provided by Professor Dr. William J. Zuercher from the University of North Carolina, USA. The PKIS is openly available to researchers to explore the biology of target proteins, and the kinase targets are well-described (Elkins et al., 2016). Therefore, the ALK inhibition by these molecules was described in the Elkins et al 2016 using Nanosyn enzyme assay. More information on the PKIS can be found at www.sgc-unc.org.
- The format of references cited by authors is inconsistent. For example, references 30, 41, 43, and so on. Although this is a small problem, it also shows the author's carelessness.
Response: Thank you for pointing this out. We corrected the format of references.
- The clarity of all the pictures in the manuscript needs to be improved.
Response: Thank you for that comment. The figures were revised to achieve a better resolution and quality.
- I would suggest to publish this paper after these questions have been addressed.
Response: We thank you for your comments and suggestions.

Reviewer 2 Report
In this research, authors have screened 147 kinase inhibitors and found only 3 molecules were significantly cytotoxic for AGP-01 cell line. And these KIs targeted on ALK in gastric cancer cell line. However, more experiments and research in real-world are needed to support this discovery.
The related researches about ALK in NSCLC are relatively mature. However, ALK gene expression in gastric carcinoma is rare. In the recent literature, ALK gene rearrangement was reported to be 8.4% (38/455) positive in gastric cancer, and is associated with younger age, signet ring cell histology and worse prognosis [25707491]. Although there were 42 cases (2.3%) of signet ring carcinoma were considered positive for ALK translocation by FISH, none of them were confirmed as positive using IHC[25755678]. Meanwhile, Glückstein and colleagues have investigated the IHC expressions of ALK in 477 adenocarcinomas of the stomach and gastroesophageal junction with negative result [PMID: 35204520]. So as Yang’s work [PMID: 26404902]. Therefore, the conclusion in this research may support a possible therapeutic direction in gastric cancer, but its’ repeatability needs further verification.
As you mentioned in Discussion, gastric carcinogenesis differs between Asian and Caucasian tumors. So, ALK expression in gastric cancer might also be different among human species and different subtypes. More researches and sub-group analysis about underlying mechanisms are wanted.
Author Response
Dear reviewer, my co-authors and I would like to thank you for the suggestions made during this high-quality review and then we present the answers to the questions.
We inform that with the reviews and suggestions, we were able to improve the idea presented by our work and we appreciate the opportunity. We hope this review has left the article suitable for publication in this high-impact journal and respect in the area.
Kind Regards.
Response to reviewer 2
In this research, authors have screened 147 kinase inhibitors and found only 3 molecules were significantly cytotoxic for AGP-01 cell line. And these KIs targeted on ALK in gastric cancer cell line. However, more experiments and research in real-world are needed to support this discovery.
The related researches about ALK in NSCLC are relatively mature. However, ALK gene expression in gastric carcinoma is rare. In the recent literature, ALK gene rearrangement was reported to be 8.4% (38/455) positive in gastric cancer, and is associated with younger age, signet ring cell histology and worse prognosis [25707491]. Although there were 42 cases (2.3%) of signet ring carcinoma were considered positive for ALK translocation by FISH, none of them were confirmed as positive using IHC[25755678]. Meanwhile, Glückstein and colleagues have investigated the IHC expressions of ALK in 477 adenocarcinomas of the stomach and gastroesophageal junction with negative result [PMID: 35204520]. So as Yang’s work [PMID: 26404902]. Therefore, the conclusion in this research may support a possible therapeutic direction in gastric cancer, but its’ repeatability needs further verification.
As you mentioned in Discussion, gastric carcinogenesis differs between Asian and Caucasian tumors. So, ALK expression in gastric cancer might also be different among human species and different subtypes. More researches and sub-group analysis about underlying mechanisms are wanted.
Response: First, studies involving the ALK gene and gastric cancer are still in progress. As you mentioned, some studies have already shown that there is alteration in the ALK gene in clinical specimens of gastrointestinal tumors. Although there are few works, our study reinforces the importance of progressing ALK studies in this type of tumor. In addition, our results with the TCGA database showed that gastric cancer patients with high ALK expression survive less than with low expression. Another important point for our work is the successful prospection of ALK inhibitors, showing their anticancer effects. Therefore, our work comes to add in this field of study little addressed in the literature and with divergences.

Round 2
Reviewer 2 Report
No more comments. I agree to accept this paper in present form.